# Parallel Tempering Monte Carlo Studies of Phase Transition of Free Boundary Planar Surfaces

**DOI:** 10.3390/polym10121360

**Published:** 2018-12-08

**Authors:** Andrey Shobukhov, Hiroshi Koibuchi

**Affiliations:** 1Faculty of Computational Mathematics and Cybernetics, Lomonosov Moscow State University, Leninskiye Gory, MSU, 2-nd Educational Building, Moscow 119991, Russia; shobukhov@cs.msu.su; 2Department of Industrial Engineering, National Institute of Technology, Ibaraki College, Nakane 866, Hitachinaka, Ibaraki 312-8508, Japan

**Keywords:** crumpling transition, graphene, graphene-based polymers, crumples, parallel tempering, Monte Carlo, statistical mechanics

## Abstract

We numerically study surface models defined on hexagonal disks with a free boundary. 2D surface models for planar surfaces have recently attracted interest due to the engineering applications of functional materials such as graphene and its composite with polymers. These 2D composite meta-materials are strongly influenced by external stimuli such as thermal fluctuations if they are sufficiently thin. For this reason, it is very interesting to study the shape stability/instability of thin 2D materials against thermal fluctuations. In this paper, we study three types of surface models including Landau-Ginzburg (LG) and Helfirch-Polyakov models defined on triangulated hexagonal disks using the parallel tempering Monte Carlo simulation technique. We find that the planar surfaces undergo a first-order transition between the smooth and crumpled phases in the LG model and continuous transitions in the other two models. The first-order transition is relatively weak compared to the transition on spherical surfaces already reported. The continuous nature of the transition is consistent with the reported results, although the transitions are stronger than that of the reported ones.

## 1. Introduction

The two-dimensional surface model proposed by Helfrich is a model for biological membranes composed of lipid molecules, and it shares almost the same mathematical structure with the Polyakov’s rigid string model for elementary particles in subatomic scales [1,2]. In those models, the extrinsic curvature or bending energy plays an essential role in maintaining the smooth shape of surfaces, and because of its mathematical transparency, many studies have been conducted on the phase structure of the model between the smooth and crumpled phases [3,4,5,6,7,8,9]. The discrete surface models defined on triangulated lattices have also been extensively studied by Monte Carlo (MC) simulations [10,11,12,13,14,15,16,17,18,19].

However, the order of the crumpling transition is still controversial. Since the discrete model depends on the discretization of continuous Hamiltonian, we have a variety of discrete models [20,21]. Indeed, the curvature energy itself has a lot of variation such as extrinsic and intrinsic curvatures. For almost all of these discrete models, MC studies predict that the models undergo a first-order crumpling transition if the lattice is of spherical topology and allowed to self-intersect (⇔ self-intersecting) [20]. Here, we should note that only self-intersecting and fixed-connectivity lattice models are studied in this paper, and self-avoiding models and fluid surface models are not considered [15,16]. Intrinsic curvature models also have a first-order crumpling transition even on a disk surface [21], and the intrinsic curvature models are also out of consideration in this paper. The first-order crumpling transition is supported by theoretical studies on the basis of non-perturbative renormalization group techniques [8,9]. In contrast, it is reported that the order of transition is of second-order on the free boundary lattices [22,23]. Therefore, this continuous transition combining the above-mentioned first-order one indicates that the order of transition depends on the surface topology; either spheres or free boundary planar disks.

In this paper, we study three different surface models by the parallel tempering MC (PTMC) technique to check whether or not the transition is of second-order on triangulated disks with a free boundary. The PTMC technique was developed to simulate the spin glass models at very low temperature, where the standard Metropolis MC technique is not effective [24,25]. It is also reported that this PTMC technique can be applied to phenomena which undergo first-order transitions [26,27]. Therefore, we expect that the PTMC technique can also be used to study the phase structure of the surface models in this paper even if these models have first-order transitions [28].

We should here comment on the reason why the crumpling transition is of interest. Indeed, graphite oxide sheets in solvents have a crumpled state [29,30]. Crumpled states can also be observed in graphenes [31,32]. The surface condition of graphenes is altered by corrugations, and therefore ripples, wrinkles and crumples emerge [33,34]. These surface states modify or enhance the material properties such as mechanical, electrical and optical ones [35]. The crumpled graphene, for example, is expected to have enhanced chemical activities and energy storage capacities [36]. In addition to pure graphenes, polymer-graphene nano-composite or graphene-based polymers (or polymer-based graphenes) also has the crumpled states [37,38,39,40]. For the application of crumpled states of these graphene-based materials, it is interesting to study their stability against thermal fluctuation or some other stimuli in environmental conditions [41]. Therefore, the crumpled state is worthwhile studying in terms of phase transition.

This paper is organized as follows: In Section 2, the three different models and the PTMC technique are described in detail. Readers who are familiar with or not interested in these topics can skip Section 2 and go to Section 3, where the simulation results including snapshots of surfaces are presented. In Section 4, we summarize the results and comment on the future study.

## 2. Models and Monte Carlo Technique

### 2.1. Triangulated Disk

Discrete surface models are defined on such a triangulated disk shown in Figure 1. The lattice is characterized by the numbers $\left(N,N_{E},N_{T}\right)$, which are the total number of vertices, the total number of edges, and the total number of triangles. Using the number *L* of division of the hexagon edge, we have the expressions for these numbers such that $\left(N,N_{E},N_{T}\right)=\left(3L^{2}+3L+1,9L^{2}+3L,6L^{2}\right)$. The lattice shown in Figure 1 is obtained by L=5, and hence $\left(N,N_{E},N_{T}\right)=\left(91,240,150\right)$. We chose a sufficiently small *L* to visualize the lattice structure. The lattices used in the simulations are larger than the lattice in Figure 1. The lattice spacing *a* for the edge length can be used as the length scale [42]. However, the simulation data are not directly compared to the experimental ones in this paper, and for this reason we fix *a* to a=1 for simplicity.

### 2.2. Landau-Ginzburg surface Model

The so-called Landau-Ginzburg surface model is first introduced and studied by Paczuski, Kardar and Nelson in [43], and this model is also numerically studied in [44]. Let $r(\in {ℜ}^{3})$ be the surface position. The continuous Hamiltonian is given by
(1)$$\begin{array}{ccc} S_{\mathrm{LG}}\left(r\right) & = & \frac{t}{2}\int d^{2}x{\left(\partial_{a}r\right)}^{2}+\frac{\kappa }{2}\int d^{2}x{\left(\partial^{2}r\right)}^{2}+u\int d^{2}x{\left(\partial_{a}r\cdot \partial_{b}r\right)}^{2}+v\int d^{2}x{\left(\partial_{a}r\cdot \partial_{a}r\right)}^{2}, \end{array}$$
where $\partial_{a}r=\partial r/\partial x_{a},\left(a=1,2\right)$ is a tangential vector along the local coordinate axis $x_{a}$ on the surface and plays a role of the order parameter. The real numbers t,κ,u,v are the coefficients of the energy terms, which are square and quadratic with respect to $\partial_{a}r$. The second term is defined by the square of second-order differentials ${\left(\partial^{2}r\right)}^{2}$.

The continuous energy $S_{\mathrm{LG}}\left(r\right)$ in Equation (Equation 1) can be written more explicitly such that
(2)$$\begin{array}{c} \begin{array}{c} S_{\mathrm{LG}}=tS_{1}+\kappa S_{2}+uS_{3}+vS_{4},\\ S_{1}=\frac{1}{2}\int d^{2}x\left[{\left(\partial_{1}r\right)}^{2}+{\left(\partial_{2}r\right)}^{2}\right],\\ S_{2}=\frac{1}{2}\int d^{2}x{\left(\partial_{1}^{2}r+\partial_{2}^{2}r\right)}^{2}=\frac{1}{2}\int d^{2}x\left(\partial_{1}^{2}r\cdot \partial_{1}^{2}r+\partial_{2}^{2}r\cdot \partial_{2}^{2}r+2\partial_{1}^{2}r\cdot \partial_{2}^{2}r\right),\\ S_{3}=\int d^{2}x\left[{\left(\partial_{1}r\cdot \partial_{1}r\right)}^{2}+{\left(\partial_{2}r\cdot \partial_{2}r\right)}^{2}+2{\left(\partial_{1}r\cdot \partial_{2}r\right)}^{2}\right],\\ S_{4}=\int d^{2}x\left[{\left(\partial_{1}r\cdot \partial_{1}r\right)}^{2}+{\left(\partial_{2}r\cdot \partial_{2}r\right)}^{2}+2{\left(\partial_{1}r\right)}^{2}{\left(\partial_{2}r\right)}^{2}\right]. \end{array} \end{array}$$

The detailed information on the role of each term is written in Ref. [44], and we briefly describe the outline of each term. The first term $S_{1}$ is given by the integration of length squares of the tangential vectors, and it simply plays a role of the tensile energy. The second term $S_{2}$ plays a role of the bending energy and the in-plane shear energy, and the third term $S_{3}$ contains both the tensile and the shear energy components. The final term $S_{4}$ is a quadratic tensile energy term.

The discrete Hamiltonian is as follows [44]:(3)$$\begin{array}{c} \begin{array}{c} S_{\mathrm{LG}}=tS_{1}+\kappa S_{2}+uS_{3}+vS_{4},\\ S_{1}=\frac{2}{3}\sum_{ij}{\left(r_{i}-r_{j}\right)}^{2}=\frac{2}{3}\sum_{i}e_{i}^{2},\\ S_{2}=\frac{1}{3}\sum_{ij}{\left(e_{i}-e_{j}\right)}^{2}+\frac{1}{3}\sum_{\left(ij),(kl\right)}\left(e_{i}-e_{j}\right)\cdot \left(e_{k}-e_{l}\right),\\ S_{3}=\frac{2}{3}\sum_{\Delta }\left[{\left(e_{1}^{2}\right)}^{2}+{\left(e_{2}^{2}\right)}^{2}+{\left(e_{3}^{2}\right)}^{2}+{\left(e_{1}\cdot e_{2}\right)}^{2}+{\left(e_{2}\cdot e_{3}\right)}^{2}+{\left(e_{3}\cdot e_{1}\right)}^{2}\right],\\ S_{4}=\frac{2}{3}\sum_{\Delta }\left[{\left(e_{1}^{2}\right)}^{2}+{\left(e_{2}^{2}\right)}^{2}+{\left(e_{3}^{2}\right)}^{2}+\left(e_{1}^{2}\right)\left(e_{2}^{2}\right)+\left(e_{2}^{2}\right)\left(e_{3}^{2}\right)+\left(e_{3}^{2}\right)\left(e_{1}^{2}\right)\right],\;\left(\mathrm{LG}\right). \end{array} \end{array}$$

We briefly summarize how to obtain the discrete Hamiltonian in Equation (Equation 3) from the continuous one in Equation (Equation 2). First of all, we note that the local coordinate origin of the triangle 123 in Figure 2a is at the vertex 1, and hence the differentials $\partial_{1}r$ and $\partial_{2}r$ in $S_{1}$ of Equation (Equation 2) are replaced by
(4)$$\begin{array}{c} \partial_{1}r\to e_{1}=r_{2}-r_{1},\;\partial_{2}r\to e_{2}=r_{3}-r_{1}. \end{array}$$

Recalling that there are two other local coordinate origins on the triangle 123, and including all possible terms for the differentials $\partial_{1}r$ and $\partial_{2}r$ and with the factor 1/3, we have
(5)$$\begin{array}{c} S_{1}=\frac{1}{3}\sum_{\Delta }\left(e_{1}^{2}+e_{2}^{2}+e_{3}^{2}\right), \end{array}$$
where $e_{3}\left(=r_{3}-r_{2}\right)$ is not written in Figure 2a. The integration $\int d^{2}x$ is replaced by the sum over triangles Δ such that $\int d^{2}x\to \sum_{\Delta }$. By replacing the summation convention from the sum over triangles $\sum_{\Delta }$ to the sum over bonds $\sum_{i}$, we have $S_{1}$ in Equation (Equation 3). We should note that this energy $S_{1}$ is also defined on the triangles of the boundary.

The second order derivative such as $\partial_{1}^{2}r$ in $S_{2}$ is replaced by $\partial_{1}^{2}r\to e_{j}-e_{i}$ using the vectors $e_{j}$ and $e_{i}$ on the hexagon in Figure 2b, because $e_{j}$ and $e_{i}$ are considered as the discretization of $\partial_{1}r$ along the coordinate axis $2^{\prime }12$ corresponding to the local coordinate axis $x_{1}$ at the vertex 1. Another diagonal line $3^{\prime }13$ on this hexagon is considered as the $x_{2}$ axis, and thereby the square of Laplacian ${\left(\partial^{2}r\right)}^{2}={\left(\partial_{1}^{2}r+\partial_{2}^{2}r\right)}^{2}$ is replaced by ${\left(e_{j}-e_{i}\right)}^{2}+{\left(e_{l}-e_{k}\right)}^{2}+2\left(e_{j}-e_{i}\right)\cdot \left(e_{l}-e_{k}\right)$ using the vectors $e_{i}$, $e_{j}$,$e_{k}$ and $e_{l}$ in Figure 2b. Thus, we have $S_{2}$ in Equation (Equation 3) as a discrete bending energy corresponding to the continuous one $\left(1/2\right)\int dx^{2}{\left(\partial^{2}r\right)}^{2}$ in Equation (Equation 3). The reason for the factor 1/3 in the discrete expression is that every vertex inside the boundary is assumed to be the center of hexagon, and therefore the summation is triply duplicated. Strictly speaking, the duplication at the vertices close to the boundary is not triple, however, the coefficient is not so important and is simply fixed to 1/3, which is the same as in the spherical model [44]. In the discrete $S_{2}$, $\sum_{ij}$ denotes the sum over the three different diagonal lines, and $\sum_{\left(ij),(kl\right)}$ denotes the sum over the corresponding local coordinates on the hexagon.

Please note that on the hexagonal lattice, such as shown in Figure 1, the coordination number at the vertices inside the boundary is given by q=6, where the coordination number $q_{i}$ is the total number of bonds emanating from the vertex *i*. This is in sharp contrast to spherical lattices, which must include the vertices with q≠6. Therefore, the vertex with q=6 in Figure 2b is sufficient for the discretization of ${\left(\partial^{2}r\right)}^{2}$ for all internal vertices. On the boundary vertices, the definition of energy $S_{2}$ is slightly different from that on the internal vertices. On the vertices with q=4, the square of Laplacian ${\left(\partial^{2}r\right)}^{2}={\left(\partial_{1}^{2}r+\partial_{2}^{2}r\right)}^{2}$ is simply replaced by ${\left(e_{j}-e_{i}\right)}^{2}$ instead of ${\left(e_{j}-e_{i}\right)}^{2}+{\left(e_{l}-e_{k}\right)}^{2}+2\left(e_{j}-e_{i}\right)\cdot \left(e_{l}-e_{k}\right)$. Moreover, the bending energy $S_{2}$ is not defined on the vertices with q=3 on the boundary (see Figure 1).

In the continuous $S_{3}$ and $S_{4}$ of Equation (Equation 2), the derivatives $\partial_{1}r$ and $\partial_{2}r$ are replaced by $e_{1}$ and $e_{2}$ in Equation (Equation 4) on the triangle in Figure 2a. The discretization technique is exactly the same as the one assumed in $S_{1}$, and hence we have the discrete energies $S_{3}$ and $S_{4}$ in Equation (Equation 3).

The partition function *Z* is given by the multiple integrations of the vertex positions such that
(6)$$\begin{array}{c} Z=\int^{\prime }\prod_{i}dr_{i}\exp\left(-S_{\mathrm{LG}}\right), \end{array}$$
where the prime in $\int^{\prime }\prod_{i}dr_{i}$ denotes that the center of the mass of surface is fixed at the origin of ${ℜ}^{3}$.

From the scale invariance of *Z*, we have
(7)$$\begin{array}{c} \langle S_{1}^{\prime }\rangle /N=3/2,\;\;S_{1}^{\prime }=tS_{1}+\kappa S_{2}+2uS_{3}+2vS_{4}, \end{array}$$
where 〈Q〉 is defined by $\langle Q\rangle =\int^{\prime }\prod_{i}dr_{i}\;Q\;\exp\left(-S_{\mathrm{LG}}\right)/Z$ [7,44]. This relation in Equation (Equation 7) can be used to check whether the simulation is correctly performed or not.

### 2.3. Canonical Model

We start with the continuous form of Hamiltonian S(r,g) of the canonical model, which is defined by a mapping r from a two-dimensional surface *M* to the three-dimensional Euclidean space ${ℜ}^{3}$, such that
(8)$$\begin{array}{c} r:M∋\left(x_{1},x_{2}\right)⟼r\left(x_{1},x_{2}\right)=\left(X,Y,Z\right)\in {ℜ}^{3}. \end{array}$$

The variable r, which is originally used to denote the surface position in ${ℜ}^{3}$, is now used for the symbol of the mapping. Another variable denoted by *g* in S(r,g) is the metric function $g_{ab}$ on *M*, where $g_{ab}$ is a 2×2 matrix. The metric $g_{ab}$ originally is not a variable but is determined by a local coordinate, which is fixed arbitrarily by hand from the reparametrization invariance. This invariance is a symmetry of the model under 2D coordinate transformations on the surface r(M) in ${ℜ}^{3}$. Thus, the model is slightly extended from the original one in the sense that $g_{ab}$ is a variable that should be physically determined. As a consequence, we have the possibility to obtain surfaces which cannot be in ${ℜ}^{3}$ in the extended model, whether this is meaningful or not. For example, let us consider the metric
(9)$$g_{ab}=\left(\begin{array}{cc} \ell_{12}^{2} & \;\ell_{12}\ell_{13}\cos\Phi \\ -\ell_{12}\ell_{13}\cos\Phi & \;\ell_{13}^{2} \end{array}\right)$$
on a surface discretized by piece-wise linear triangles such as in Figure 3a, where cosΦ is not always identical to $\cos\phi ={\vec{\ell }}_{12}\cdot {\vec{\ell }}_{13}/\ell_{12}\ell_{13}$ between the edge vectors ${\vec{\ell }}_{12}$ and ${\vec{\ell }}_{13}$ [45]. If Φ=ϕ for all triangles, then this metric is identical with the discrete induced metric $g_{ab}=e_{a}\cdot e_{b}$. However, if the angles {Φ} do not satisfy the triangle equality, i.e., the sum of three internal angles is not always equal to π, then the surface with such metric in Equation (Equation 9) generally cannot be realized in ${ℜ}^{3}$.

The continuous Hamiltonian is given by
(10)$$\begin{array}{c} S\left(r,g\right)=\gamma \int_{M}\sqrt{g}\;d^{2}x\;g^{ab}\partial_{a}r\cdot \partial_{b}r+\frac{\kappa }{2}\int_{M}\sqrt{g}\;d^{2}x\;g^{ab}\partial_{a}n\cdot \partial_{b}n,\;\left(\gamma =1\right) \end{array}$$
where the surface tension coefficient γ is always fixed to γ=1. The symbol $g^{ab}$ denotes the inverse of $g_{ab}$, and *g* is its determinant. We should note that $g_{ab}$ assumed in the expression of S(r,g) in Equation (Equation 10) is a variable that should be determined just like the mapping r as mentioned above [46]. Note also that this Hamiltonian is a two-dimensional extension of the polymer model of Doi-Edwards [47].

Here, we should note that the real surface r(M) in ${ℜ}^{3}$ corresponding to the material under consideration is described by the induced metric $g_{ab}=\partial_{a}r\cdot \partial_{b}r$. This allows us to consider that the surface with a given metric $g_{ab}$ is different from the real surface, pointing to the possibility for the surface to correspond to this $g_{ab}$. Therefore, from this set of surfaces, a physically meaningful surface should be uniquely determined by the modeling. For this reason, we introduce a two-dimensional surface *M* in addition to the real surface with $\partial_{a}r\cdot \partial_{b}r$, both of which should be physically determined. This is another extension of the surface model, and this is the meaning of the mapping described in Equation (Equation 8). Please note that the surface *M* is not necessarily a manifold [45,48,49].

The problem is then how to determine $g_{ab}$ for *M*. One possible and simple technique is to fix $g_{ab}$ to the Euclidean metric such that $g_{ab}=\delta_{ab}$. In this case, *M* is a simple two-dimensional Euclidean space and plays no role in describing the model, however, we use *M* to express the domain in the two-dimensional integrations in S(r). Thus, the only variable to be determined is r, and S(r) is now given by
(11)$$\begin{array}{c} S\left(r\right)=\gamma \int_{M}\;d^{2}x\;\partial_{a}r\cdot \partial_{a}r+\frac{\kappa }{2}\int_{M}\;d^{2}x\;\partial_{a}n\cdot \partial_{a}n,\;\left(\gamma =1\right). \end{array}$$

One simple reason for why we assume the Euclidean metric $\delta_{ab}$ for $g_{ab}$ is as follows: The Hamiltonian in Equation (Equation 10) is invariant under an arbitrary conformal transformation $g_{ab}\to g_{ab}^{\prime }=f\left(x\right)g_{ab}$ with a positive function f(x) on *M*. This invariance is described by $S\left(r,g\right)=S\left(r,g^{\prime }\right)$ and implies that the metric $g_{ab}$ can be chosen relatively freely, and therefore it is fixed to the simplest one such that $g_{ab}=\delta_{ab}$.

The discrete Hamiltonians are obtained from the continuous one in Equation (Equation 11) by the replacement of the differentials in Equation (Equation 4). Here, we use the symbols ${\vec{\ell }}_{12}=e_{1}$ and ${\vec{\ell }}_{13}=e_{2}$ for the edge vectors (Figure 3a), and $\ell_{ij}=|{\vec{\ell }}_{ij}\left|=\right|r_{j}-r_{i}|$ for the edge (or bond) length. Thus, we obtain
(12)$$\begin{array}{c} S\left(r\right)=S_{1}+\kappa S_{2},\\ S_{1}=\sum_{\left(i,j\right)}{\left(r_{i}-r_{j}\right)}^{2}=\sum_{\left(i,j\right)}\ell_{ij}^{2},\;S_{2}=\sum_{\left(i,j\right)}\left(1-n_{i}\cdot n_{j}\right),\;\left(\mathrm{cano}\right), \end{array}$$
where the factor 4/3 is eliminated from both $S_{1}$ and $S_{2}$ for simplicity. This $S_{1}$ is exactly the same as $S_{1}$ of the LG model in Equation (Equation 3) up to the numerical factor. The symbol $\sum_{\left(ij\right)}$ in $S_{1}$ denotes the sum over all bonds (ij) connecting the vertices *i* and *j*. In contrast, (ij) in the sum $\sum_{\left(ij\right)}$ of $S_{2}$ denotes the triangles sharing a common bond (Figure 3b), and the unit normal vector $n_{i}$ is defined on the triangle *i*. The partition function of the canonical model is exactly the same as *Z* in Equation (Equation 6) for the LG model except the Hamiltonian in the Boltzmann factor.

### 2.4. Modified Canonical Model

The third model, which we call “modified canonical model”, is obtained from the same continuous Hamiltonian in Equation (Equation 11) assumed for the canonical model. The only difference is a discretization of the bending energy $S_{2}$, where the unit normal vector n(i) at the vertex *i* (Figure 4a) is used as well as the normal vector $n_{i}$ on the triangle *i* (Figure 3b). The discrete Hamiltonian is given by [20]
(13)$$\begin{array}{c} S\left(r\right)=S_{1}+\kappa S_{2},\\ S_{1}=\sum_{\left(i,j\right)}\ell_{ij}^{2},\;S_{2}=\sum_{i=1}^{N}\sum_{j(i)}\left[1-n\left(i\right)\cdot n_{j(i)}\right],\;\left(\mathrm{modi}\right), \end{array}$$
where $n_{j(i)}$ in $S_{2}$ is the unit normal vector of the triangle j(i) connected to the vertex *i*. The normal vector n(i) at the vertex *i* is defined by (Figure 4a)
(14)$$n\left(i\right)=N_{i}/\left|N_{i}\right|,\;N_{i}=\sum_{j(i)}n_{j(i)}A_{j(i)},$$
where $A_{j(i)}$ is the area of the triangle j(i). Please note that the interaction range of the normal vectors $n_{i}$ is slightly larger than that of the canonical model (Figure 4b). In fact, only two nearest neighbor vectors $n_{i}$ and $n_{j}$ are directly coupled in $S_{2}$ in Equation (Equation 12) of the canonical model, and as a consequence only three nearest neighbor vectors $n_{i}\left(i=1,2,3\right)$ are coupled to $n_{0}$ (see Figure 3b). In contrast, as shown in Figure 4b, the non-nearest neighbor $n_{i}$ and $n_{j}$, of which the triangles *i* and *j* do not directly contact each other, are coupled to $n_{0}$ in $S_{2}$ in Equation (Equation 13) of the modified model.

### 2.5. Parallel Tempering Monte Carlo Technique

The so-called parallel tempering Monte Carlo (PTMC) technique developed for the spin glass model at low temperatures is successfully applied to the first-order crumpling transition of the canonical model on spherical lattices [28]. In this subsection, the outline of the PTMC technique applied to the tethered surface model is briefly presented.

Let (r,κ) represent a system of configuration $r(=\left\{r_{1},r_{2},\cdots ,r_{N}\right\})$ with a given κ, which is assumed to be changed. In the case of the LG model, the parameter κ is also changed while the other three parameters (t,u,v) are fixed in the simulations in this paper. In this PTMC, $N_{R}$ replicas $\left\{\left(r_{1},\kappa_{1}\right),\left(r_{2},\kappa_{2}\right),\cdots ,\left(r_{N_{R}},\kappa_{N_{R}}\right)\right\}$ are simulated in parallel by the standard Metropolis MC (MMC) simulation technique [50,51], and the systems are exchanged after sufficiently long MMC runs. Because of this exchange, the total number of different combinations $\left\{\left(r_{1},\kappa_{1}\right),\left(r_{2},\kappa_{2}\right),\cdots ,\left(r_{N_{R}},\kappa_{N_{R}}\right)\right\}$ is $N_{R}!$.

The $N_{R}$ replicas are updated as follows:(P_1_)Perform long MMC simulations for $N_{R}$ replicas(P_2_)Exchange all nearest neighbor systems (r,κ) and $\left(r^{\prime },\kappa^{\prime }\right)$ with the probability
(15)$$\begin{array}{c} W\left(r,\kappa |r^{\prime },\kappa^{\prime }\right)=\mathrm{Min}\left[1,\exp(-\Delta )\right],\;\Delta =\left(\kappa^{\prime }-\kappa \right)\left[S_{2}\left(r\right)-S_{2}\left(r^{\prime }\right)\right] \end{array}$$(P_3_)Repeat $P_{1}$ and $P_{2}$

We should note that the process $P_{1}$ can be performed in parallel as mentioned above. In the exchange process $P_{2}$, only bending energy $S_{2}$ is used, and no information on the other energy $S_{1}$ is used in the canonical and modified canonical models. This is also true for the LG model, and no information on the energies $S_{1}$, $S_{3}$ and $S_{4}$ is left out in the exchange process.

Figure 5a,b intuitively show the difference of MMC and PTMC simulations for four replica systems. The numbers 1,2,3,4 denote the bending rigidities such as $\kappa_{1},\cdots ,\kappa_{4}$, and the color blocks denote the configurations $r_{1},\cdots ,r_{4}$. The combination of $\left(r_{i},\kappa_{j}\right)$ is exchanged as the iterations evolve in the PTMC simulation.

Here, we briefly show that all micro states {r,κ} satisfy the canonical Boltzmann distribution as a result of PTMC simulations. Let P({r,κ}) be a probability distribution for all the states {r,κ} such that
(16)$$\begin{array}{c} P\left(\left\{r,\kappa \right\}\right)=\prod_{m=1}^{N_{R}}P_{\mathrm{eq}}\left(r_{m},\kappa_{m}\right),\;P_{\mathrm{eq}}\left(r,\kappa \right)=Z^{-1}\exp\left[-S(r,\kappa )\right],\;S\left(r,\kappa \right)=\bar{S}\left(r\right)+\kappa S_{2}\left(r\right), \end{array}$$
where $\bar{S}\left(r\right)$ is independent of κ and given by $\bar{S}\left(r\right)=tS_{1}\left(r\right)+uS_{3}\left(r\right)+vS_{4}\left(r\right)$ for the LG model and $\bar{S}\left(r\right)=S_{1}\left(r\right)$ for the other two models. The $P_{\mathrm{eq}}\left(r,\kappa \right)$ in Equation (Equation 16) is the Boltzmann distribution function of the state (r,κ). If the PTMC exchange satisfies the detailed balance condition described by
(17)$$\begin{array}{c} P\left(\ldots ,;r,\kappa ;\ldots ;r^{\prime },\kappa^{\prime };\ldots \right)W\left(r,\kappa |r^{\prime },\kappa^{\prime }\right)=P\left(\ldots ,;r^{\prime },\kappa ;\ldots ;r,\kappa^{\prime };\ldots \right)W\left(r^{\prime },\kappa |r,\kappa^{\prime }\right), \end{array}$$
then we understand from the well-known uniqueness theorem that this *P* is the uniquely determined probability [28]. Please note that the right-hand side of Equation (Equation 17) is obtained from the left-hand side by exchanging r and $r^{\prime }$. Thus, we should prove that P({r,κ}) in Equation (Equation 16) satisfies the condition in Equation (Equation 17). This can be accomplished in two steps. The first step is to see that Equation (Equation 17) for P({r,κ}) in Equation (Equation 16) is equivalent with the relation
(18)$$\begin{array}{c} \frac{W(r,\kappa |r^{\prime },\kappa^{\prime })}{W(r^{\prime },\kappa |r,\kappa^{\prime })}=\exp\left(-\Delta \right) \end{array}$$
under the condition given by Equation (Equation 15). The second step is to see that the relation in Equation (Equation 18) is correct. This second step is almost trivial from the definitions of $W(r,\kappa |r^{\prime },\kappa^{\prime })$ and Δ in Equation (Equation 15). The first step is also straightforward to prove.

The assumed parameters for the simulations including the total number of MC sweeps (MCS) are listed in Table 1, where 1 MCS consists of the processes $P_{1}$ and $P_{2}$ of PTMC. In Table 1, the symbol *N* is the total number of vertices, #total (MCS) and #therm (MCS) are the total number of MCS and the total number of thermalization MCS, and $n_{P_{1}}$ denotes the number of iterations performed in $P_{1}$ per 1 MC sweep. This $n_{P_{1}}$ is fixed to $n_{P_{1}}=10$ (or $n_{P_{1}}=20$), and this implies that the total number of MMC (including thermalization) iterations for each replica is 10 (or 20) times larger than the #total (MCS). The $N_{R}$ is the total number of replicas, and $\kappa_{1}$ and $\kappa_{N_{R}}$ ($\kappa_{1}<\kappa_{N_{R}}$) are the bending rigidity of the replica 1 and $N_{R}$, $\Delta \kappa (=\left(\kappa_{N_{R}}-\kappa_{1}\right)/N_{R})$ is the difference of κ between two neighboring replicas, which will be exchanged in the process $P_{2}$. The total number of iterations #total for the large lattices in the latter two models is not so large compared to those for smaller lattice in the LG model.

## 3. Simulation Results

### 3.1. Snapshots

Firstly, in the presentation section, we show snapshots, which are obtained as one of the configurations from the replicas i=1 and $i=N_{R}$ in each model. In the upper (lower) low in Figure 6, the snapshots are obtained from the replica $i=N_{R}$ (i=1) of the (a) LG, (b) canonical, and (c) modified canonical models. The size of lattices are the largest for these snapshots, and hence from Table 1 all $N_{R}$ are given by $N_{R}=24$. From Table 1, we find that the values of κ corresponding to these replicas are given as follows: (a) κ=0.1835 (upper), κ=0.187 (lower), (b) κ=0.766 (upper), κ=0.782 (lower), and (c) κ=0.451 (upper), κ=0.467 (lower). In the LG model simulations, the parameters (t,u,v) are fixed to
(19)$$\begin{array}{c} (t,u,v)=(-6,0.2,0.2),\;(\mathrm{LG}). \end{array}$$

We find that the surfaces in the upper low are globally bending but almost flat while those in the lower low shrink to a small ball. From this observation, we understand that the surfaces in the upper and lower lows are in the flat and crumpled phases, respectively, in all of the models.

### 3.2. Bending Energy and Mean Square Gyration

The bending energy of the LG model is calculated only on the internal vertices, because the definition of $S_{2}$ on the boundary vertices is slightly different from that on the internal vertices as described in Section 2.2. In Figure 7a–i, we plot the bending energy $S_{2}/N_{B}$ of the LG model and the other models, the specific heat
(20)$$\begin{array}{c} C_{S_{2}}=\frac{\kappa^{2}}{N}\langle {\left(S_{2}-\langle S_{2}\rangle \right)}^{2}\rangle , \end{array}$$
and the peak $C_{S_{2}}^{\max}$ of the specific heat. We should note that $N_{B}$ is the total number of internal bonds $N_{B}=3\left(N-6L\right)$ for the LG model, where N−6L is the total number of internal vertices and hence 3(N−6L) is the total number of the diagonal lines such as $2^{\prime }12$ in Figure 2b. The reason why this $N_{B}$ is used for $S_{2}$ in the LG model is because the first term of $S_{2}$ in Equation (Equation 3) is defined on such diagonal lines, and the second term is also defined on a pair of diagonal lines $2^{\prime }12$ and $3^{\prime }13$ in Figure 2b, and there are three different such diagonal lines on each internal vertex. For the other two models, $N_{B}\left(=N_{E}-6L\right)$ is the total number of internal bonds, on which the bending energy $S_{2}$ is defined, where $N_{E}$ is the total number of edges $N_{E}$ given in Section 2.1. Please note that $C_{S_{2}}$ in Equation (Equation 24) for the LG model is defined also by using *N*. The curves of $C_{S_{2}}$ in Figure 7h for the modified model are fluctuating, and this fluctuation may be due to the fact that the total number of iterations is not sufficient for this model as mentioned above, though reasonable peak values $C_{S_{2}}^{\max}$ are obtained.

From the LG model data plotted in Figure 7a–c, we see that the $S_{2}/N_{B}$ abruptly changes against κ, and correspondingly the $C_{S_{2}}$ has the anomalous peak showing the existence of the crumpling transition. As mentioned above, the parameters (t,u,v) are fixed to the values in Equation (Equation 19) and remain unchanged. The peak heights $C_{S_{2}}^{\max}$ vs. *N* are plotted in log-log scale in Figure 7c, where *N* is used. The data obtained by the other models are also plotted in Figure 7d–i. Fitting the largest three data of $C_{S_{2}}^{\max}$ to the scaling relation
(21)$$\begin{array}{c} C_{S_{2}}^{\max}∼N^{\sigma }, \end{array}$$
we have
(22)$$\begin{array}{c} \sigma =1.96\pm 0.07\;(\mathrm{LG}),\;\sigma =0.72\pm 0.01\;(\mathrm{cano}),\;\sigma =0.79\pm 0.05\;(\mathrm{modi}). \end{array}$$

We find from these results that the transition of the LG model is of the first order while in the other two models the transition is of second-order, because σ≥1 in the LG model and σ<1 in the other models. We note that σ should be σ=1 for the first order transitions, because $C_{S_{2}}^{\max}$ is expected to scale as $L^{2}\left(=N\right)$, where *L* is the linear size of the lattice [52]. However, the present result is almost twice of this expectation and indicates that the dimension *D* of lattice is effectively D=4. One possible reason is that the surface is self-intersecting, and hence, the total number or density of possible configurations is larger than that in the case of D=2. The first result in Equation (Equation 22) of the LG model is qualitatively comparable to σ=1.58±0.08 of the canonical model on spherical lattice in Ref. [28], because both results indicate that the transition is of first-order. In contrast, the latter two results in Equation (Equation 22) of the other models indicate a second-order transition, and hence these two are completely different from the result in Ref. [28]. We consider that this difference simply stems from the difference in the lattice topology or structure; sphere and disk, which are surfaces without a boundary and with a free boundary, or are compact and non-compact.

We calculate the coefficient σ in Equation (Equation 21) for the LG model using the specific heat $C_{S_{2}}$ of the bending energy $S_{2}$ in Equation (Equation 12), and we have σ=1.96±0.07, which is almost identical to the first result σ=1.97±0.07 in Equation (Equation 22). This indicates that the results σ=1.96±0.07 in Equation (Equation 22) for the LG model is reliable, although this result is obtained from the bending energy on the internal vertices only.

The problem here is ascertaining why the result of LG model in Equation (Equation 22) is different from the others. Firstly, we should be reminded that it is predicted that the LG model has a continuous crumpling or wrinkling transition from the mean filed analysis [11]. However, as mentioned in Ref. [28], the basic assumption for the mean field analysis that the surface fluctuation is relatively small is not always true even for the models. This is considered to be why the result of the LG model deviates from the mean field prediction and has the first-order transition even on the lattice with a free boundary. Next, we have to consider where the difference comes from in the order of transitions between the LG model and the other two models. One possible origin is the shear stress or resistance to shear deformation expected in the LG model; it is not expected in the other models. The bending energy $S_{2}$ assumed in the LG models is identical to $S_{2}$ in the other models up to a numerical factor, and this $S_{2}$ has no shear resistance to in-plane deformation of triangles; it has only a resistance to the bending or out-of-plane deformation. Thus, the only source for the shear resistance is the Gaussian bond potential $S_{1}$ in the two models, while in the LG model, $S_{3}$ and $S_{4}$ as well as $S_{1}$ have resistance to shear stresses. Here, we should note that all models are fixed-connectivity or tethered models, and the shear resistance is not negligible compared to the fluid surface models on dynamically triangulated lattices. In the tethered models, any deformations of triangles except simple expansion/shrinkage accompany a shear resistance, because all triangles tend to become regular in the equilibrium configurations. Please note that the simple expansion and shrinkage of triangles are suppressed because the mean bond length remains constant due to the scale invariant-property of the partition function.

The mean square radius of gyration $R_{g}^{2}$ defined by
(23)$$\begin{array}{c} R_{g}^{2}=\frac{1}{N}\sum_{i}{\left(r_{i}-\bar{r}\right)}^{2},\;\bar{r}=\frac{1}{N}\sum_{i}r_{i}, \end{array}$$
its variance
(24)$$\begin{array}{c} C_{R_{g}}=\frac{1}{N}\langle {\left(R_{g}^{2}-\langle R_{g}^{2}\rangle \right)}^{2}\rangle , \end{array}$$
and the peak values $C_{R_{g}}^{\max}$ are plotted in Figure 8a–i. The fluctuations of the data $C_{R_{g}}$ in Figure 8h are relatively large due to the same reason for $C_{S_{2}}$ mentioned above. We also find that $R_{g}^{2}$ rapidly changes, where $C_{R_{g}}$ has a peak $C_{R_{g}}^{\max}$. The peak position on the κ axis for $C_{R_{g}}^{\max}$ of the LG model is almost identical with the position for $C_{S_{2}}^{\max}$, while those for $C_{R_{g}}^{\max}$ and $C_{S_{2}}^{\max}$ in the other models are considerably different from each other. This difference is not observed in the canonical model on spherical lattice in Ref. [28], where a first-order transition is expected and where the transition point is very clear and detected uniquely even by numerical simulations. In contrast, continuous transitions are relatively unclear in general, and hence the transition points of the latter two models are relatively hard to observe. Moreover, the total number of iterations for these are not always sufficiently large as mentioned above. These are possible reasons for the deviation of the transition points observed in $C_{R_{g}}^{\max}$ and $C_{S_{2}}^{\max}$.

The peak values $C_{R_{g}}^{\max}$ are expected to scale according to
(25)$$\begin{array}{c} C_{R_{g}}^{\max}∼N^{\mu }. \end{array}$$

By fitting the largest three data of $C_{R_{g}}^{\max}$ in Figure 8c,f,i to this relation, we have
(26)$$\begin{array}{c} \mu =1.73\pm 0.08\;(\mathrm{LG}),\;\mu =0.93\pm 0.12\;(\mathrm{cano}),\;\mu =0.89\pm 0.08\;(\mathrm{modi}). \end{array}$$

These results also support the proposition that the LG model has a first-order transition and the other models a second-order transition, though the coefficient of the canonical model is close to μ=1 and the transition is close to a first-order one. The reason why μ in Equation (Equation 26) of the LG model is μ>1 may be the fact that the surface is not self-avoiding [28]. We should note that the order of transition depends on its definition. If we call a transition first-order only if the coefficient μ for the variance $C_{S_{*}}$ of Hamiltonian $S_{*}$ is μ=1, then the canonical and modified models clearly have a second-order transition from the results σ in Equation (Equation 22). In general, the coefficient σ for the bending energy $S_{2}$ is more important than μ as a coefficient for the determination of the order of transition. Thus, it is reasonable to consider that the crumpling transitions of the canonical and modified canonical models are continuous transitions.

### 3.3. Binder Quantity and Fractal Dimension

Firstly. in this subsection, we calculate the Binder quantity $B_{S_{2}}$ defined by [53,54]
(27)$$B_{S_{2}}=1-\frac{\langle {\left(S_{2}-\langle S_{2}\rangle \right)}^{4}\rangle }{3{\langle {\left(S_{2}-\langle S_{2}\rangle \right)}^{2}\rangle }^{2}}$$
for the LG model (Figure 9a). It is expected that $B_{S_{2}}$ has a peak $B_{S_{2}}^{\max}$ and $B_{S_{2}}^{\max}\to 2/3$ at the first-order transition point. To see whether this expectation is satisfied or not, $B_{S_{2}}^{\max}$ vs. 1/N are plotted in Figure 9b, where the Binder quantity $B_{R_{g}}^{\max}$ for $R_{g}^{2}$ is also plotted. The solid lines are also drawn by Mathematica command “Interpolation” for the data of $B_{S_{2}}^{\max}$ and $B_{R_{g}}^{\max}$. The value $B_{S_{2}}^{\max}\left(N\to \infty \right)$ on the vertical axis expected from the extrapolations is almost identical with 2/3, while $B_{R_{g}}^{\max}\left(N\to \infty \right)$ is slightly smaller than 2/3. This deviation of $B_{R_{g}}^{\max}\left(N\to \infty \right)$ seems due to the size effect, because the lattice size is not so large compared to those used in Ref. [28], where $B_{S_{2}}^{\max}\left(N\to \infty \right)≃0.7$ and $B_{R_{g}}^{\max}\left(N\to \infty \right)≃0.69$ are observed.

A first-order transition is also reflected in the Binder cumulant $V_{S_{2}}$, which is defined by [53,54]
(28)$$V_{S_{2}}=1-\langle S_{2}^{4}\rangle /\left(3{\langle S_{2}^{2}\rangle }^{2}\right).$$

The Binder cumulant $V_{R_{g}}$ for $R_{g}^{2}$ is also defined analogously to Equation (Equation 28). These quantities are expected to have the minimum $V_{S_{2}}^{\min}$ and $V_{R_{g}}^{\min}$ at the first-order transition point, and this expectation is confirmed in Figure 9c, where only $V_{S_{2}}^{\min}$ is plotted. It is also found that the position of $V_{S_{2}}^{\min}$ on the κ axis is almost the same as that of $C_{S_{2}}^{\max}$ in Figure 7b. This also implies that $V_{S_{2}}^{\min}$ reflects the first-order transition.

In the other two models, the quantities $B_{S_{2}}$ and $V_{S_{2}}$ (and those for $R_{g}^{2}$) are unclear compared to the case of LG model. One of the reasons for this is the continuous nature of the transition; the convergent speed of the simulation is very slow close to the transition point.

Next, we calculate the fractal dimension $D_{f}$, which is defined by
(29)$$\begin{array}{c} R_{g}^{2}∼N^{2/D_{f}}. \end{array}$$

To calculate this quantity at the transition point, we plot $R_{g}^{2}$ vs. *N* in Figure 9d–f in the log-log scale, where $R_{g}^{2}$ are obtained at the peak position of $C_{R_{g}}$ in Figure 8b,e,h. By fitting the plotted data to Equation (Equation 29), we obtain
(30)$$\begin{array}{c} D_{f}=2.34\pm 1.01\;\left(\mathrm{LG}\right),\;D_{f}=2.54\pm 0.29\;\left(\mathrm{cano}\right),\;D_{f}=2.36\pm 0.60\;\left(\mathrm{modi}\right). \end{array}$$

Since the errors of $R_{g}^{2}$ are large in the first and third models (see Figure 9d,f), the errors in $D_{f}$ are also relatively large. The values of $D_{f}$ within these errors are comparable to $D_{f}=2.50\pm 0.30$ at the first-order transition point of the canonical model on the spherical lattice in Ref. [28].

To check the simulations are performed correctly, we can use the relations
(31)$$\begin{array}{c} S_{1}^{\prime }/N=3/2\;\left(\mathrm{LG}\right),\\ S_{1}/N=3/2\;\left(\mathrm{cano},\;\mathrm{modi}\right), \end{array}$$
where $S_{1}^{\prime }$ of the first one is given in Equation (Equation 7), and $S_{1}^{\prime }=tS_{1}+\kappa S_{2}+2uS_{3}+2vS_{4}$. The second equality is also obtained by using almost the same technique for the first one. The symbol 〈·〉 is omitted for these $S_{1}^{\prime }$ and $S_{1}$ for simplicity. The results plotted in Figure 9g–i are consistent with these predictions in Equation (Equation 31).

## 4. Summary and Conclusions

In this paper, we study the crumpling transition of a planar surface with a free boundary by parallel tempering Monte Carlo (PTMC) simulations using three different tethered lattice models; the Landau-Ginzburg (LG) model, the canonical (cano) model, and the modified canonical (modi) model, defined on triangulated fixed-connectivity lattices of disk topology. The order of the transition is first order in the LG model, while it is of second order in the other models. This second-order nature of the transition is consistent with the result reported in [22,23].

The models studied in this paper are not self-avoiding, however, the transition between the crumpled and smooth phases indicates that these states are stable against thermal fluctuations. This can give insights into studies on real materials such as graphene, where the crumpled states are expected to have many technological applications.

In the presence of impurity, a new phase is expected to appear, and the surface critical exponents are also expected to change from those of the model without impurity [55]. Thus, it is of great interest to study the planar surface model with impurities.

## Figures and Tables

**Figure 1 polymers-10-01360-f001:**
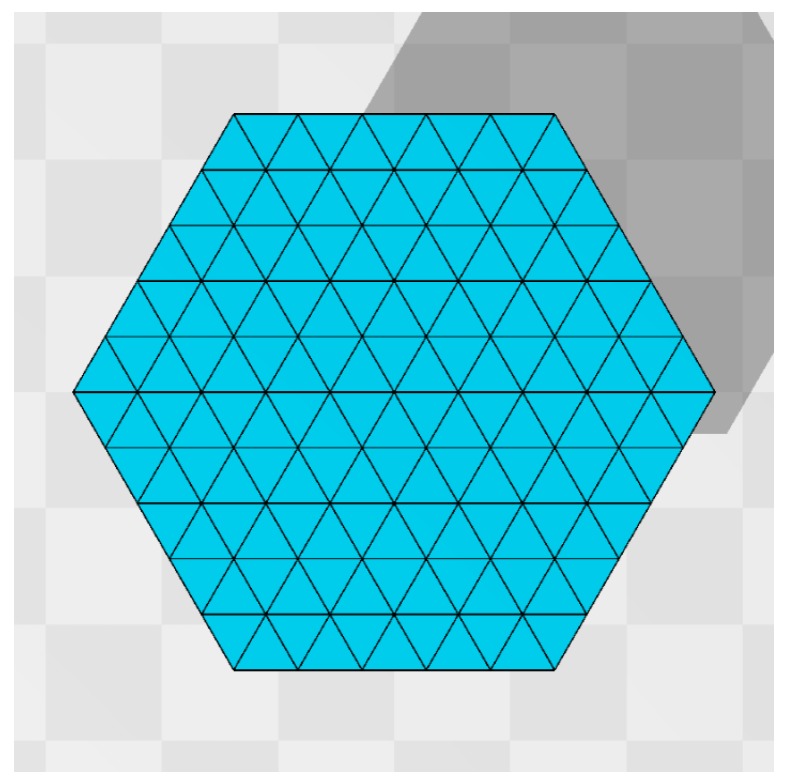
A hexagonal lattice discretized by regular triangles. The total number of vertices *N* including those on the boundary is given by N=91. This number is calculated by the formula $N=3L^{2}+3L+1$, where L (=5) is the number of division of the edge of the original hexagon. This type of hexagonal lattice is used to define discrete Hamiltonians of the surface models studied in this paper.

**Figure 2 polymers-10-01360-f002:**
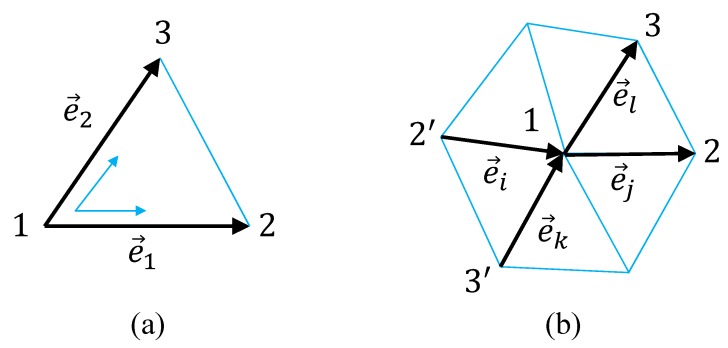
(**a**) A local coordinate on a triangle 123 and the edge vectors $e_{1}$ and $e_{2}$ along the coordinates axes $x_{1}$ and $x_{2}$. A local coordinate for the discretization of the second-order differential $\partial^{2}r$ in $S_{2}$ of LG model on (**b**) a vertex of coordination number q=6 (hexagonal).

**Figure 3 polymers-10-01360-f003:**
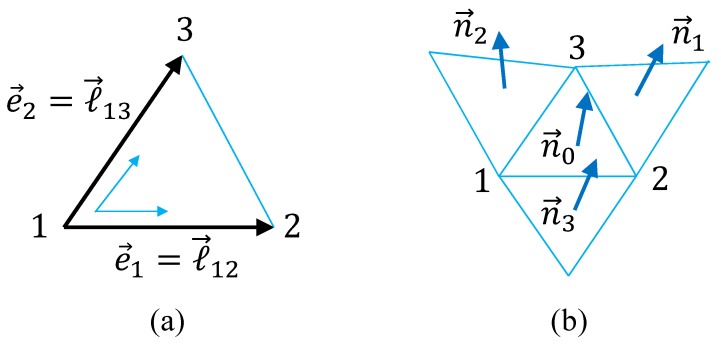
Lattice structure for discretization of the canonical surface model. (**a**) A local coordinate on the triangle 123 with the edge vectors ${\vec{\ell }}_{12}$ and ${\vec{\ell }}_{13}$, and (**b**) the triangle 123 and its three nearest-neighbor triangles, where the normal vector $n_{0}$ interacts with $n_{i}\left(i=1,2,3\right)$. The local coordinate in (**a**) is exactly the same as that in Figure 2a.

**Figure 4 polymers-10-01360-f004:**
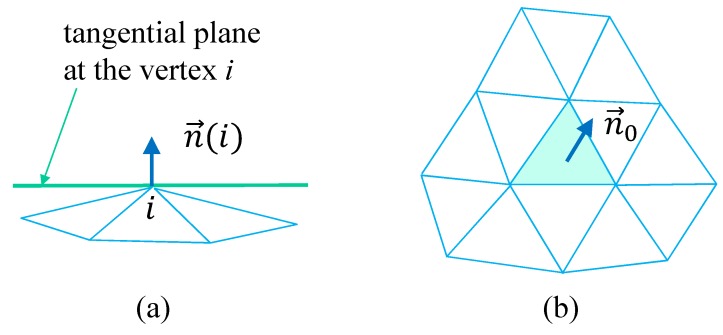
Lattice structure for discretization of the modified canonical surface model. (**a**) The tangential plane at the vertex *i* and its normal vector n(i), and (**b**) a triangle and its neighboring triangles, where the normal vector $n_{0}$ interacts with those of the neighboring triangles. The range of interaction is slightly larger than that shown in Figure 3b.

**Figure 5 polymers-10-01360-f005:**
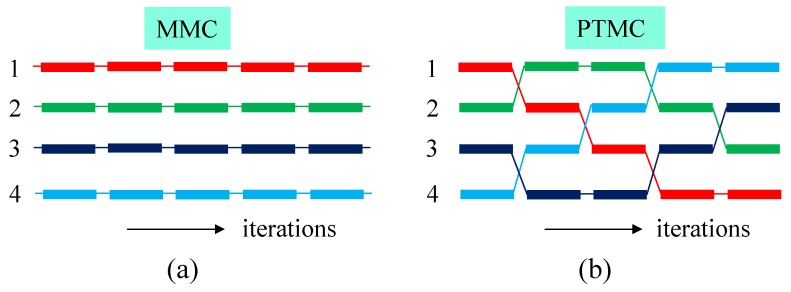
Illustration of how the replica systems evolve in (**a**) MMC and (**b**) PTMC simulations. In (**a**) MMC simulations, each system evolves independent of the other systems, while in (**b**) PTMC simulations, the systems are exchanged.

**Figure 6 polymers-10-01360-f006:**
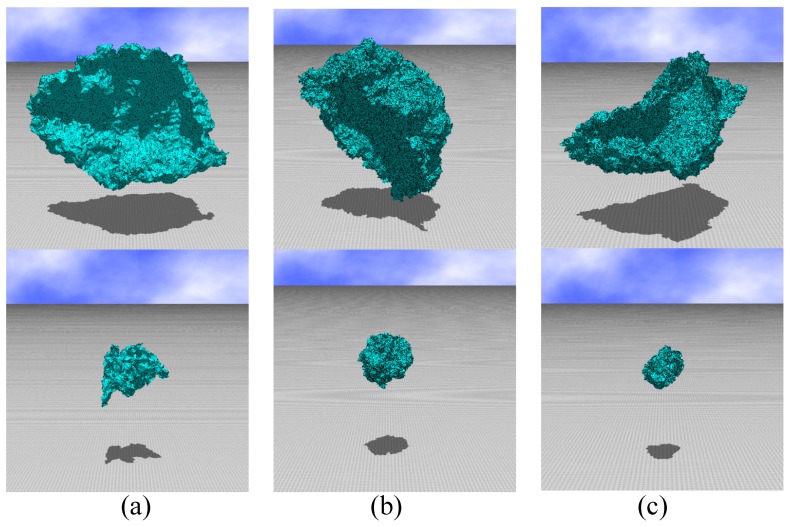
Snapshots of surfaces obtained in the replica $i(\in \left\{1,\cdots ,N_{R}\right\})$ such that $i=N_{R}\left(=24\right)$ in the upper low and i=1 in the lower low, which correspond to the flat phase and the crumpled phase, respectively, in each model. The models and lattice size *N* are (**a**) the LG model, N=7351, (**b**) the canonical model, *N* = 44,287, and (**c**) the modified canonical model, *N* = 20,917.

**Figure 7 polymers-10-01360-f007:**
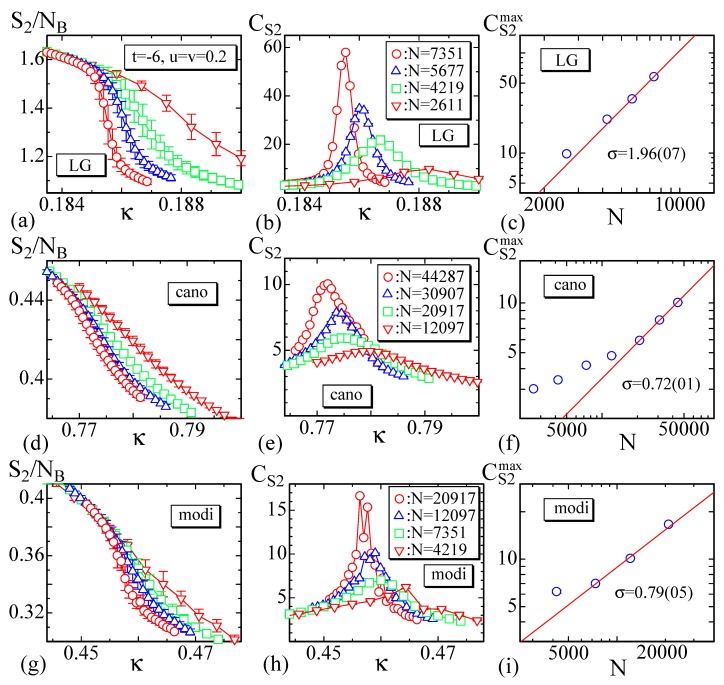
The bending energy per bond $S_{1}/N_{B}$, the specific heat $C_{S_{2}}$, and the log-log plot of the peak value $C_{S_{2}}^{\max}$ vs. *N* for (**a**–**c**) the LG model, (**d**–**f**) the canonical model, and (**g**–**i**) the modified canonical model.

**Figure 8 polymers-10-01360-f008:**
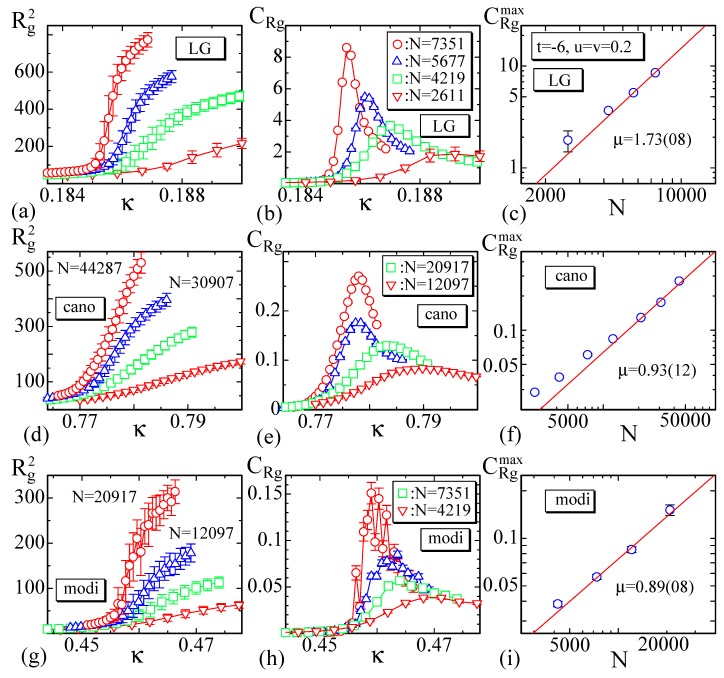
The mean square gyration $R_{g}^{2}$, its variance $C_{R_{g}}$, and the log-log plot of the peak value $C_{R_{g}}^{\max}$ vs. *N* for (**a**–**c**) the LG model, (**d**–**f**) the canonical model, and (**g**–**i**) the modified canonical model.

**Figure 9 polymers-10-01360-f009:**
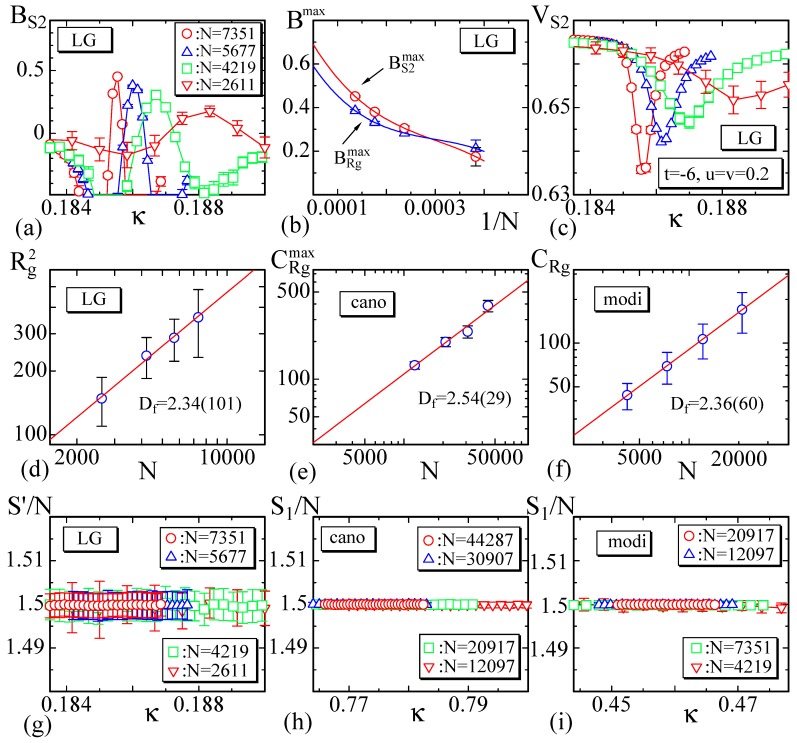
(**a**) The Binder quantity $B_{S_{2}}$, (**b**) the peak values $B_{S_{2}}^{\max}$ and $B_{R_{g}}^{\max}$ vs 1/N with solid lines drawn by Mathematica command “Interpolation”, and (**c**) the Binder cumulant $V_{S_{2}}$ for the LG model. The log-log plot of $R_{g}^{2}$ vs. N of (**d**) the LG model, (**e**) the canonical model, and (**f**) the modified canonical model. $S^{\prime }$ of (**g**) the LG model, and $S_{1}/N$ of (**h**) the canonical model and (**i**) the modified canonical model.

**Table 1 polymers-10-01360-t001:** The parameters assumed for the simulations; $n_{P_{1}}$ denotes the total number of MMC iterations performed in the $P_{1}$ process per 1 MCS for each replica, $\kappa_{1}$ and $\kappa_{N_{R}}$ ($\kappa_{1}<\kappa_{N_{R}}$) are the bending rigidity of the replica 1 and $N_{R}$, and $\Delta \kappa (=\left(\kappa_{N_{R}}-\kappa_{1}\right)/N_{R})$ is the difference of κ between two neighboring replicas.

Model	*N*	#Total (MCS)	#Therm (MCS)	$n_{P_{1}}$	$N_{R}$	$\kappa_{1}$	$\kappa_{N_{R}}$	Δκ
LG	7351	$2.5\times 10^{8}$	$2.5\times 10^{7}$	10	24	0.1835	0.187	$1.46\times 10^{-4}$
LG	5677	$2.5\times 10^{8}$	$2.5\times 10^{7}$	10	24	0.1842	0.1878	$1.5\times 10^{-4}$
LG	4219	$9\times 10^{7}$	$3\times 10^{7}$	10	24	0.183	0.194	$4.58\times 10^{-4}$
LG	2611	$1\times 10^{7}$	$2\times 10^{6}$	10	24	0.18	0.2	$8.33\times 10^{-4}$
cano	44,287	$3.8\times 10^{7}$	$1.8\times 10^{7}$	20	24	0.766	0.782	$2.5\times 10^{-4}$
cano	30,907	$8\times 10^{7}$	$3\times 10^{7}$	20	24	0.764	0.787	$9.58\times 10^{-4}$
cano	20,917	$1\times 10^{8}$	$1.5\times 10^{7}$	20	24	0.762	0.792	$1.25\times 10^{-3}$
cano	12,097	$7.5\times 10^{7}$	$7.5\times 10^{6}$	10	24	0.77	0.804	$1.42\times 10^{-3}$
modi	20,917	$6.4\times 10^{7}$	$1\times 10^{7}$	20	24	0.451	0.467	$6.67\times 10^{-4}$
modi	12,097	$1.9\times 10^{8}$	$2.5\times 10^{7}$	10	24	0.448	0.47	$9.167\times 10^{-4}$
modi	7351	$5\times 10^{8}$	$5\times 10^{7}$	10	16	0.444	0.476	$2\times 10^{-2}$
modi	4219	$5\times 10^{8}$	$1.5\times 10^{7}$	10	16	0.44	0.49	$6.96\times 10^{-2}$

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
