# Peer review of "Parallel Tempering Monte Carlo Studies of Phase Transition of Free Boundary Planar Surfaces"

_polymers, 2018, doi:10.3390/polym10121360_

Round 1

Reviewer 1 Report

Parallel Tempering Monte Carlo Studies of Phase
Transition of Free Boundary Planar Surfaces
by Andrey Shobukhov and Hiroshi Koibuchi

The authors of this paper studied the transition between the rumpled phase and the smooth  phase of a triangulated membrane with free boundary and disk topology.
They have used the Parallel Tempering MC Simulation method to calculate physical properties
from which they deduced some finite-size scaling exponents.  
Calculations have been carried out for three models: LG, canonical, modified canonical.
From the results they concluded that the transition in the LG model is of first order, while that in the other two models is of second order.

My remarks are the following:

1. The descriptions of the discrete Hamiltonian and the PTMC procedure used in this paper are long but very clear. This may help pedagogically unfamiliar researchers in this field easily follow the remaining of the paper.  

2. The authors did not explain why for first order transitions one \sigma>1.  In lattice theory for phase transitions, physical quantities scale as L^d (system volume) where d is the system dimension and L the linear size.  If I put roughly N=L^2 in the membrane and d=2, then the authors obtained for LG model
Cmax_S2= N^\sigma=L^\sigma/2 \simeq L^0.98 (with \sigma=1.96).

Cmax thus does not scale as the system volume. Can the authors explain more about this point?

3. There are errors in the caption of Fig. 7. There are errors of notation in Fig. 8 (in the last figure in each row, \nu and \sigma should be \mu).

Point 2 is important because it is the main result of the paper. I recommend the authors to explain this point before the paper can be recommended for publication.

Author Response

Please see the attached file "Shobukhov-Koibuchi-response-to-the-referees.pdf".

Reviewer 2 Report

The manuscript entitled "Parallel tempering Monte Carlo studies of phase transition of free boundary planar surfaces" is devoted to an actual and quite highlighted topic which fits well the scope of the journal. The obtained results are clearly described; the objectives of the work are clearly justified with respect to the state-of-the-art; used references are fully appropriate. The impact of the obtained results within the specific field is high. Conclusions are valuable and fully supported by the gained results.  Hence, the manuscript should be of definite interest for the readers of the journal. I expect the manuscript will be appropriately cited if published.

Minor corrections of the English language and style is required. The manuscript can be considered for publication after a minor "language" revision.

Author Response

(The authors gave the same response as above.)

Round 2

Reviewer 1 Report

In this revised version, the authors have added a paragraph (p. 11)  to reply my main question (remark 2 in my previous report). I think that it is satisfactory. They have also corrected a number of errors in the figures.

I think that the paper is now suitable for publication.